# Enhancement of Macrophage Immunity against Chlamydial Infection by Natural Killer T Cells

**DOI:** 10.3390/cells13020133

**Published:** 2024-01-11

**Authors:** Ying Peng, Sai Qiao, Hong Wang, Sudhanshu Shekhar, Shuhe Wang, Jie Yang, Yijun Fan, Xi Yang

**Affiliations:** 1Department of Immunology, Rady Max College of Medicine, University of Manitoba, Winnipeg, MB R3E 0T5, Canada; 2Department of Medical Microbiology, School of Medicine, Shandong University, Jinan 250100, China

**Keywords:** *Chlamydia*, macrophage, iNKT cells, M1, JAK/STAT signalling pathway

## Abstract

Lung macrophage (LM) is vital in host defence against bacterial infections. However, the influence of other innate immune cells on its function, including the polarisation of different subpopulations, remains poorly understood. This study examined the polarisation of LM subpopulations (monocytes/undifferentiated macrophages (Mo/Mφ), interstitial macrophages (IM), and alveolar macrophages (AM)). We further assessed the effect of invariant natural killer T cells (iNKT) on LM polarisation in a protective function against *Chlamydia muridarum*, an obligate intracellular bacterium, and respiratory tract infection. We found a preferentially increased local Mo/Mφ and IMs with a significant shift to a type-1 macrophage (M1) phenotype and higher expression of iNOS and TNF-α. Interestingly, during the same infection, the alteration of macrophage subpopulations and the shift towards M1 was much less in iNKT KO mice. More importantly, functional testing by adoptively transferring LMs isolated from iNKT KO mice (iNKT KO-Mφ) conferred less protection than those isolated from wild-type mice (WT-Mφ). Further analyses showed significantly reduced gene expression of the JAK/STAT signalling pathway molecules in iNKT KO-Mφ. The data show an important role of iNKT in promoting LM polarisation to the M1 direction, which is functionally relevant to host defence against a human intracellular bacterial infection. The alteration of JAK/STAT signalling molecule gene expression in iNKT KO-Mφ suggests the modulating effect of iNKT is likely through the JAK/STAT pathway.

## 1. Introduction

*Chlamydia* is an obligate intracellular bacterial pathogen causing multiple health problems in humans. *C. pneumoniae* causes respiratory diseases such as bronchitis, sinusitis, and pneumonia, while *C. trachomatis* is a major cause of ocular and genital tract sexually transmitted diseases [1,2]. Most efforts to develop a safe and rational chlamydial vaccine have been hindered by the lack of adequate understanding of the protective and pathologic immune responses to *Chlamydial* infection [3,4]. Therefore, in recent decades, significant efforts have been made to study the immunological basis of protection and pathology in *chlamydial* diseases. Notably, significant progress has been made recently in chlamydia vaccine development [5,6,7], but further study on the immune cells and cell interactions is needed. In the experimental studies, *Chlamydia muridarum* (*C. m*.), a mouse strain of Chlamydia, has been widely used using mouse models of respiratory and genital tract *chlamydial* infections [8].

The macrophage is one of the most important immune cells involving innate and adaptive immunity and plays a critical role in sensing and controlling bacterial infections [9,10]. All the macrophages express F4/80 on their surface in the lung [11]. There are mainly two distinct macrophage subpopulations in the lung at homeostasis: alveolar (AMs) and interstitial (IMs). AMs, located in the airway lumen, are characterised by high expression of CD11c. Conversely, IMs reside in the lung parenchyma, expressing low levels of CD11c. AMs play an essential role in maintaining immunological homeostasis in the lung, whereas IMs are thought to have a regulatory function within the lung tissue [12,13]. In addition, some monocytes and undifferentiated macrophages (Mo/Mφ), which lack CD11c expression in the lung, can differentiate into macrophages during inflammation and infection [14,15]. The function of the macrophage subpopulations is related to their polarisation status. Specifically, the polarised status of macrophages has been broadly defined as type-1 (M1) and type-2 (M2) macrophages, based on their molecular signature and function, analogous to the T helper cells’ Th1/Th2 paradigm. M1 macrophages produce higher pro-inflammatory cytokines and molecules (iNOS and TNF-α) and promote local Th1 responses, while M2 macrophages predominantly produce anti-inflammatory cytokines and play a role in type-2 immune responses [16,17,18]. Four major signalling pathways (JNK pathway, JAK/STAT pathway, PI3K/Akt pathway, and Notch pathway) are involved in M1 polarisation [19,20]. In vitro, M1 macrophages restricted the growth of *C. trachomatis*, while M2 macrophages were more suitable to promote this [21,22]. Our previous studies have shown the critical role of nitric oxide (NO) production by macrophages in inhibiting *chlamydial* infection [23] and the detrimental effect of macrophage depletion or apoptosis in *chlamydial* lung infection [24]. More recently, we found that NK cells promote M1 polarisation through up-regulation of miR-155 in macrophages by producing IFN-γ during *chlamydial* infection [25].

Natural killer T (iNKT) cells represent a subset of innate lymphocytes that possess the characteristics of NK and αβ T cells. iNKT cells play an essential role in the early stages of immune responses and can also produce cytokines that regulate subsequent adaptive immune responses against infectious pathogens [26,27]. Our previous work examined the interactions of iNKT/NK and iNKT/DC after Chlamydia lung infection and found that iNKT is a critical regulator in innate and adaptive immune responses [28,29,30]. However, the impact of iNKT cells on macrophages in *chlamydial* infections has yet to be studied. In this study, we investigated the polarisation status of each macrophage subpopulation in *C. m*. lung infection and examined the influence of iNKT cells on macrophage polarisation during in vivo *C. m*. infection. The results show a significant polarisation of macrophages in *chlamydial* lung infection and a great impact of iNKT cells on the shift of macrophage patterns during the infection.

## 2. Materials and Methods

### 2.1. Mice

Wild-type C57BL/6 mice (WT) and Jα18 KO mice (iNKT KO) were bred at the University of Manitoba animal care facility. Breeding pairs of Jα18 KO (iNKT KO) mice in the B6 background were initially kindly provided by Dr. Masaru Taniguchi (RIKEN Research Center for Allergy and Immunology, Yokohama, Japan). iNKT KO mice were generated by specific deletion of the Jα18 gene segment using homologous recombination and aggregation chimera techniques [31]. All mice used in this study were males of 6–8 weeks of age. All experiments complied with the institutional guidelines issued by the Canadian Council of Animal Care.

### 2.2. Organism

*C. muridarum* organisms (Nigg strain) were cultured, purified, and quantified as previously described [32]. Briefly, *C. muridarum* was grown in HeLa 229 cells (ATCC, Burlington, ON, Canada) in RPMI 1640 medium supplemented with 10% fetal bovine serum (FBS), 1% L-glutamine, and 25 mg/mL gentamycin (Thermo Fisher, Mississauga, ON, Canada). The elementary bodies (EBs) were purified by discontinuous density gradient centrifugation. The infectivity of purified EBs was measured by infecting Hela 229 and immunostaining *chlamydial* inclusions. The purified EBs were suspended in sucrose–phosphate–glutamic acid (SPG) buffer (Sigma-Aldrich, Oakville, ON, Canada) and stored at −80 °C. The same batch of purified EBs was used throughout this study.

### 2.3. Mouse Infection Cell Transfer and Quantitation of In Vivo Bacterial Loads

Mice were intranasally inoculated with 1 × 10^3^ inclusion-forming units (IFU) of *C. muridarum* EBs in 40 μL SPG buffer. The mice were killed at specified time points after infection, and the lungs were collected to analyse immune cells, cytokines, and bacterial burden. In adoptive transfer experiments, lung F4/80^+^ macrophages from WT or iNKT KO mice isolated using a MACS F4/80 column (Miltenyi Biotec, San Diego, CA, USA) were first washed in protein-free PBS and then intranasally delivered into naive WT-recipient mice (5 × 10^5^ cells/mouse). Two hours after adoptive transfer, the mice were inoculated with 1 × 10^3^ IFUs of *C. m*. in 40 μL of PBS. The mice’s body weights were recorded daily before (day 0) and after inoculation. The mice were killed at a specified time, and the lungs were aseptically isolated and processed to assess *C. m.* in vivo growth quantitatively. The lungs were aseptically isolated and mechanically homogenised at predetermined days after inoculation using a cell grinder in SPG buffer. The tissue homogenates were centrifuged at 1900× *g* for 30 min at 4 °C to remove coarse tissues and debris, and the supernatants were stored frozen at −80 °C until tested. For *C. m.* quantitation, 100 μL of serially diluted organ tissue supernatants were inoculated onto Hela 229 cells grown to confluence in 96-well flat-bottom microtiter plates. The plates were centrifuged at 1100× *g*, then incubated at 37 °C for another 30 min. The cell layers were then washed, and 200 μL of MEM containing cycloheximide (2 μg/mL), gentamicin (10 μg/mL), and vancomycin (25 μg/mL) (Thermo Fisher, Mississauga, ON, Canada) were added to each well. The plates were incubated for 72 h at 37 °C in 5% CO_2_. After incubation, the culture medium was removed and cell monolayers were fixed with methanol (Sigma-Aldrich, Oakville, ON, Canada), stained with Chlamydia-specific murine mAb and HRP-conjugated goat anti-mouse IgG secondary Abs (Thermo Fisher, Mississauga, ON, Canada), and developed with the substrate (4-chloro-1-naphthol; Sigma-Aldrich, Oakville, ON, Canada). The number of inclusions was counted under a microscope at ×100 magnification. Five fields through the midline of each well were counted. *C. m*. titers in the organs were calculated based on dilution titers of the original inoculum [33].

### 2.4. Analysis of Lung Pathology

The mouse lung tissues were perfused with PBS and fixed in 10% formalin. Tissue sections were routinely stained with H&E (hematoxylin and eosin) (Sigma-Aldrich, Oakville, ON, Canada) and examined by light microscopy. The degree of lung inflammation and destruction was scored using a semi-quantitative grading system: 0, normal; 1, mild and limited inflammation, granuloma formation, cellular infiltration in less than 25% of the area, and no apparent infiltration into adjacent alveolar septa or air space; 2, mild interstitial pneumonitis, diffused cellular infiltration in some area (25–50%), septal congestion, and interstitial edema; 3, inflammatory cell infiltration into perivascular, peri-bronchiolar, alveolar septa, and air space (50–75% of the area); 4, over 75% of the area of the lung filled with infiltrating cells.

### 2.5. Flow Cytometry Analysis

Lung cells were isolated for flow cytometry analysis as described [24]. Briefly, lung tissues were harvested from mice at specified time points post-infection and digested in 2 mg/mL collagenase XI (Sigma-Aldrich, Oakville, ON, Canada) in RPMI 1640 for 1 h at 37 °C. After digestion, the cell suspension was filtered, and RBCs were removed by ACK lysis buffer (150 mM NH_4_Cl, 10 mM KHCO_3_, and 0.1 mM EDTA) (Thermo Fisher, Mississauga, ON, Canada). All the cells were washed and resuspended in a FACS buffer (Dulbecco’s PBS without Ca^2+^ and Mg^2+^ containing 2% FBS and 0.09% NaN_3_) (eBioscience, San Diego, CA, USA) and were blocked with anti-CD16/CD32 Abs (eBioscience, San Diego, CA, USA) in FACS buffer for 20 min, followed by surface staining with anti-CD45-FITC, anti-F4/80-PEcy7, anti-CD11c-Allophycocyanin, and anti-CD206-BV605 mAbs (eBioscience, San Diego, CA, USA). After being fixed and washed in permeabilisation buffer, cells were stained with anti-iNOS-PE, with anti-TNF-α-PE mAbs (eBioscience, San Diego, CA, USA), or with corresponding isotype control Abs for 30 min. Cells were washed twice with permeabilisation buffer and analysed by flow cytometry. Numbers of various cell types were identified using F4/80^pos^CD11c^high^ for alveolar macrophages (AMs), F4/80^pos^CD11c^low^ for interstitial macrophages (IMs), and F4/80^pos^CD11c^neg^ for monocytes and undifferentiated macrophages (Mo/Mφ) [34]. CD206 was used to detect M2 macrophages, and iNOS was used to identify M1 macrophages [35]. All the data were collected using an LSR II flow cytometer (BD Biosciences, San Diego, CA, USA) and analysed using Flowjo 7.6.1 (Flowjo LLC, Ashland, OR, USA).

### 2.6. Macrophages (F4/80^+^ Cells) Isolation

Mice were killed on day 3 following intranasal *C. m.* inoculation, and the lungs were aseptically removed and processed into single-cell suspensions. Lung cells were blocked with anti-CD16/CD32 Abs on ice for 15 min. For selecting F4/80^+^ cells, the suspensions were stained with PE-anti-F4/80 mAbs (eBioscience, San Diego, CA, USA), followed by anti-PE IgG1-magnetic microbeads as described by the manufacturer (Miltenyi Biotec, San Diego, CA, USA). Magnetic sorting was performed on a VarioMACS with an LS column (Miltenyi Biotec, San Diego, CA, USA). After the first selection, the F4/80^+^ cells were reloaded twice on the column to improve purity. The purity of the sorted macrophages (F4/80^+^ cells) was more than 95% based on flow cytometry analysis (Appendix A).

### 2.7. Quantitative Real-Time Analysis for Signalling Molecules in LMs

WT and iNKT KO mice were treated with *C. m*. (1 × 10^3^ IFUs/mouse) and sacrificed at days 0 and 3 after *C. m*. infection. Lungs were isolated, and the LMs were separated by MACS. The total RNA of LMs was extracted using TRIzol reagent according to the manufacturer’s instructions (Invitrogen/Life Technologies, Burlington, ON, Canada). Nanodrop measured the OD260/280 of extracted RNA, and all showed OD260/280 within the range of 1.9 to 2.0. A constant amount of RNA (1.5 μg per sample) was reverse transcribed into cDNA by using a High-Capacity cDNA Reverse Transcription Kit (ThermoFisher, Mississauga, ON, Canada), and the cDNA was then amplified using murine-specific primers. Primer sequences were as follows: JAK1 (129 bp), forward: 5′-CTGTCTACTCCATGAGCCAGCT-3′, reverse: 5′-CCTCATCCTTGTAGTCCAGCAG-3′; JAK2 (147 bp), forward: 5′-GCTACCAGATGGAAACTGTGCG-3′, reverse: 5′-GCCTCTGTAATGTTGGTGAGATC-3′; PI3K (150 bp), forward: 5′-ACCATCAGTGGCTCTGCGGTTT-3′, reverse: 5′-GTGGTCTTCTGGGAACTCACCT-3′; STAT1 (141 bp), forward: 5′-GCCTCTCATTGTCACCGAAGAAC-3′, reverse: 5′-TGGCTGACGTTGGAGATCACCA-3′; JNK (150 bp), forward: 5′-CGCCTTATGTGGTGACTCGCTA-3′, reverse: 5′-GATCAATATGATCTGTACCTGG-3′; NF-TC (131 bp), forward: 5′-GCTGCCAAAGAAGGACACGACA-3′, reverse: 5′-GGCAGGCTATTGCTCATCACAG-3′; SOCS1 (216 bp), forward: 5′-CTGCGGCTTCTATTGGGGAC-3′, reverse: 5′-AAAAGGCAGTCGAAGGTCTCG-3′; Notch1 (126 bp), forward: 5′-GCTGCCTCTTTGATGGCTTCGA-3′, reverse: 5′-CACATTCGGCACTGTTACAGCC-3′. Housekeeping gene glyceraldhyde-3-phosphate dehydrogenase (GAPDH) served as the internal control. Primers for GAPDH are forward primer 5′-CAATGTGTCCGTCGTGGAT-3′ and reverse primer 5′-AGCCCAAGATGCCCTTCAG-3′. A total of 40 cycles were used, and each cycle included denaturation (95 °C, 15 s), annealing (60 °C, 30 s), and extension (72 °C, 45 s). Real-time quantitative PCR was performed with an ABI 7500 Real-Time PCR System and analysed with 7500 System SDS software version 1.3.1 (Applied Biosystems, Foster City, CA, USA), following manufacturer’s instructions.

### 2.8. Statistical Analysis

All statistical analyses were performed using GraphPad’s Prism software (version 9.0). Statistical significance was determined using a two-way analysis of variance (ANOVA) with Sidak’s multiple comparisons. Cell counts, measured by flow cytometry, were determined for each cell type, and each cell type was calculated as a percentage of the total live-cell population. The analysis of cell-type quantification was performed using two-way ANOVA with Sidak’s multiple-comparison test. For qPCR, the molecules were measured by the 2^−ΔΔCt^ method, and the data are shown as the ratio of each gene to the housekeeping gene. Statistical significance for each group was determined using a nonparametric Mann–Whitney test. Statistical significance was determined using one-way ANOVA with repeated measurements for the bodyweight curve. For histological scoring and chlamydial IFU detection, statistical significance was determined using a Kruskal–Wallis test with Dunn’s post-test. *n* values for each experiment are included in the figure legends. Data are presented as mean ± SD. Values of *p* < 0.05 were considered significant (* *p* < 0.05; ** *p* < 0.01; *** *p* < 0.001; **** *p* < 0.0001). All the experiments were repeated two or three times with similar results.

## 3. Results

### 3.1. C. m. Infection Induces an Increase in Macrophages in the Lung

F4/80 is a well characterised and extensively referenced mouse macrophage marker in the lung within the CD45^+^ cell population [11]. We, therefore, first examined F4/80^+^ cell responses following *C. m.* lung infection in WT and iNKT KO mice. Flow cytometry assessed the changes in F4/80^+^ cells at days 0, 3, 7, and 14 post-infection. As shown in Figure 1, myeloid cells were gated by CD45 staining after excluding doublets and debris (Figure 1A). The percentage and the absolute number of macrophages increase quickly in the lung, peaking at day 7 post-infection, followed by a fast drop to baseline at day 14 post-infection (Figure 1B,C). This pattern is in line with our previous reports on the general inflammation in the lung following *C. m.* infection [8]. Although both WT and iNKT KO mice showed the same trend of macrophage response, the degree of macrophage increase in the lung was significantly lower in iNKT KO mice than in WT mice. The results suggest that *C. m*. lung infection can induce strong macrophage responses, while iNKT cells can influence this process.

### 3.2. The Changes in Macrophage Subpopulations in WT or iNKT KO Mice after the C. m. Challenge

After showing the general increase in macrophages in the lung following *C. m.* infection, we further examined the impact of the infection on the subpopulations of macrophages following the infection. To explore the role of iNKT cells in making potential changes, we also compared WT and iNKT KO mice for their macrophage subpopulations following infection. A representative flow cytometry analysis of the three macrophage subpopulations (Mo/Mφ, IMs, and AMs) is shown in Figure 2A,B. Notably, in the WT mice, the percentage of IMs increased from 12.9% on day 0 to 37.3% on day 3 and further increased to 62.4% on day 7, while AMs decreased from 57.8% on day 0 to 7.54% on day 3 and further decreased to 6.66% on day 7 post-infection. On day 14 post-infection, 19.2% of macrophages were IMs, and 49.3% were AMs. These kinetics data suggest that *C. m*. infection preferentially promotes IM while reducing AM. However, in the iNKT KO mice, while the same trend is seen (increased IM and reduced AM following infection), the degree of changes was much less significant (Figure 2A,B). There was no significant difference in Mo/Mφ percentages between iNKT KO mice and WT mice during the infection (Figure 2A,B). However, when the absolute number of each population was analysed, it was found that IM and Mo/Mφ increased significantly post-infection in WT mice, although the IM increased more dramatically. Again, in the iNKT KO mice, the increase in these two subpopulations was much less compared to WT mice. Interestingly, the absolute number of AM was not significantly changed during the infection, and the lack of iNKT cells did not impact AM (Figure 2C). Therefore, the relative reduction in AM percentage shown in Figure 2B was primarily a result of the increase in IM and Mo/Mφ. The results show a preferential promoting effect of iNKT cells on the expansion of IM and Mo/Mφ following *C. m*. lung infection.

### 3.3. Polarisation Status of Mo/Mφ, IMs, and AMs in WT and iNKT KO Mice Following C. m. Lung Infection

The polarisation of macrophage subpopulations towards M1or M2 is connected to their function in inflammation and infection [35]. To examine the functional changes in macrophages following a *chlamydial* lung infection, we explored the polarisation status of the macrophage subpopulations (AMs, IMs, and Mo/Mφ) at early and later stages of infection using iNOS and CD206 as respective M1 and M2 macrophage markers (Figure 3, Figure 4 and Figure 5). At the same time, the influence of iNKT cells on macrophage subpolarisations in the infection was also examined by comparing WT and iNKT KO mice. In general, we found that the frequency and the absolute number of iNOS^+^ macrophages in all three subpopulations were increased significantly following infection in WT mice, suggesting a significant M1 switch caused by infection, but iNKT KO mice showed an apparent reduction in this M1 switch (Figure 3, Figure 4 and Figure 5). The impact of the iNKT cell on M1 polarisation of the three macrophage subpopulations appeared to have slightly different kinetics. Specifically, it was observed that iNKT KO mice exhibited less M1-like Mo/Mφ (Figure 3) and AM (Figure 5) on day 3 but less M1-like IM on day 7 post-infection (Figure 4).

In contrast, the absolute number of CD206^+^ macrophages (M2-like) in all the subpopulations had no significant difference between WT and iNKT KO mice at the tested various time points, representing early and late stages of infection. Consequently, the ratio of M1/M2 increased significantly following infection, especially in the WT mice. Moreover, contrary to the minimal impact of infection on the number of AMs (Figure 2C), a switch of AM towards M1 was observed following infection. In addition, the iNKT KO mice showed reduced M1-like AM compared to WT mice (Figure 5), suggesting a role of iNKT cells in promoting M1 polarisation of AM as well. The results suggest that *C. m*. lung infection induces polarisation of all the tested macrophage subpopulations towards M1, during which iNKT cells significantly promote.

### 3.4. iNKT Depletion Reduces TNF-α Production of IMs and Mo/Mφ In Vivo

In addition to the differences in surface markers and enzymes, M1 and M2 macrophage subsets also have different patterns of cytokine production. In particular, tumour necrosis factor α (TNF-α) is a typical cytokine produced by M1 macrophages and contributes to cell function in inflammation and infections [36,37,38]. To further explore the effect of iNKT cells on macrophage subsets in *chlamydial* infection, we examined the amount of TNF-α protein in the macrophage subpopulations (AM, IM, and Mo/Mφ) of WT and iNKT KO mice on day 7 post-infection by intracellular cytokine staining. As shown in Figure 6, compared with WT mice, we saw a marked decrease in TNF-α-producing cells in the IM and Mo/Mφ populations of the iNKT KO mice following *chlamydial* lung infection. The data demonstrate that iNKT cells enhance TNF-α production by macrophages, especially IM and Mo/Mφ.

### 3.5. Adoptive Transfer of Macrophages from WT Mice Generates Significantly More Robust Protection than from iNKT KO Mice

The significantly higher expression of iNOS and TNF-α by macrophages from WT mice than iNKT KO mice indicates an apparent promoting effect of iNKT cells on M1 polarisation in *chlamydial* lung infection. To confirm whether these phenotypic changes reflect macrophage functions in host defence, we further used the adoptive transfer approach to directly compare the functional capacity of WT-Mφ and iNKT KO-Mφ to reduce infection and control diseases. We intranasally transferred WT-Mφ or iNKT KO-Mφ to syngeneic naive recipient mice and then challenged them with *C. m.* lung infection. Mice receiving PBS alone before the same challenge infection were used as controls. We found that WT-Mφ is significantly more protective than iNKT KO-Mφ, although both WT-Mφ and iNKT KO-Mφ are protective compared to PBS. Specifically, the bodyweight loss of the mice receiving WT-Mφ was much milder than that of the iNKT KO-Mφ recipients (Figure 7A).

Moreover, the histopathological analysis showed that the WT-Mφ recipients had significantly less inflammation and lower pathological scores at all the tested time points (days 7, 14, and 21) post-infection than the recipients of iNKT KO-Mφ (Figure 7B,C). More importantly, the *chlamydial* load in the lungs of WT-Mφ recipients was significantly lower than those of the iNKT KO-Mφ recipients (Figure 7D). Collectively, these results indicated that WT-Mφ is more potent in conferring protective immunity than iNKT KO-Mφ, confirming the role of iNKT cells in enhancing the function of macrophages against *chlamydial* lung infection.

### 3.6. iNKT Cells Promote M1 Polarisation by Activating the JAK/STAT Signalling Pathway

After showing the enhancing capacity of iNKT cells to M1 polarisation in *chlamydial* infection, we further explored the signalling pathways related to the M1 promoting effect. Figure 8A summarises the major molecules involving the four signalling pathways that promote macrophage polarisation towards the M1 phenotype [39]. We isolated LMs from infected WT and iNKT KO mice at various time points post-infection and measured the mRNA levels of the selected signalling molecules typically related to each of the four signalling pathways by real-time PCR (Figure 8B). For the JNK pathway, the expression of JNK mRNA showed no difference between WT-Mφ and iNKT KO-Mφ. For the JAK/STAT pathway, the mRNA levels of JAK1, JAK2, PI3K, and STAT1 in the iNKT KO-Mφ were significantly lower than WT-Mφ.

Regarding the PI3K/Akt pathway, the mRNA levels of NF-κB and SOCS1 were similar between the two groups, although the PI3K mRNA level was lower in iNKT KO-Mφ, which is likely due to its overlap with the JAK/STAT pathway. The mRNA levels of Notch1 and NF-κB in the Notch pathway were also similar between WT-Mφ and iNKT KO-Mφ. The JAK/STAT pathway was the only signalling pathway deficient in the iNKT KO-Mφ. There were no significant differences between the two groups of macrophages in other signalling pathways, including the JNK pathway, PI3K/Akt pathway, and Notch1 pathway. Since IFN-γ is a vital cytokine involving the JAK/STAT pathway activation, we further measured the IFN-γ levels in the lung homogenates of WT mice and iNKT KO mice on days 0, 3, 7, and 14 post-infection (Figure 8C). Consistently, IFN-γ levels in the lung homogenates of WT mice were much higher than those of iNKT KO mice, even in the very early stage (day 3 p.i.) of infection. The results suggest that iNKT cells may promote M1 polarisation by activating the JAK/STAT pathway during *C. m*. infection.

## 4. Discussion

The importance of tissue macrophage subsets in polarisation to M1- or M2-like cells has recently drawn significant attention because of their functional difference in infections and inflammations [40]. This study characterised the dynamic changes in three subpopulations of macrophages (Mo/Mφ, IMs, and AMs) in the lung following intranasal *C. m.* infection. We found dramatic changes in the total number of macrophages and the relative percentage of the different subpopulations in the early and later stages of the disease. In particular, the IM, a relatively small macrophage population in the homeostatic situation, quickly became the dominant population in the early stages of infection and reduced to baseline level in the late/resolving stage of the disease. In addition, all three subpopulations showed polarisation towards an iNOS-expressing M1-like direction during the infection process, especially for the IM population.

Interestingly, we found that iNKT cells play a critical role in promoting the expansion of macrophages in the lung. More importantly, the polarisation of all three examined subpopulations towards the M1-like direction is likely through the JAK/STAT signalling pathway. Furthermore, using adoptive transfer experiments, we confirmed the role of iNKT cells on macrophage function in controlling *chlamydial* infection by comparing macrophages from WT and iNKT KO mice. These data demonstrate that macrophages are one of the major players in host defence against a clinically relevant intracellular bacterial infection, in which iNKT cells can significantly modulate these different subpopulations in their expansion and function. Although the CD206-expressing M2-like cells did not show significant changes in iNKT KO mice compared to WT mice, the higher M1-like population increased the ratio of M1/M2, thus impacting macrophage function in host defence.

An interesting finding in this study is the dramatic increase in the lung IM population following *chlamydial* infection. For the three studied macrophage subpopulations, AM is considered an ancient cellular compartment of innate immunity found in the lung, where it plays an essential role in lung homeostasis [38]. IMs are thought to have a regulatory function within lung tissues and have a greater propensity to release specific cytokines associated with adaptive immune response [12,22]. Our data show that most of the macrophages are AM (60%) before *C. m*. infection in the mouse lung, while IM is only a small proportion of the total macrophage group (10%). However, the pattern quickly reversed after infection; IM became the predominant population on day 7 post-infection while AM reduced to the smallest population (Figure 2). Not only for its relative proportion, but the absolute number of AMs also showed a mild decrease during *C. m*. infection.

In contrast, IM increased more than 20 times on day 7 post-infection (Figure 2). The dynamic change in macrophage subpopulations is consistent with the idea that AM mainly maintains lung homeostasis at uninfected status while IMs play a critical role in controlling the infection. Monocytes and undifferentiated macrophages can transform into mature macrophages during inflammation [41]; thus, they likely contribute to increased IM in the lung following infection. Therefore, the mild increase (onefold) in Mo/Mφ may not reflect the general recruitment of monocytes and undifferentiated macrophages into the lung during the infection. Mo/Mφ AMs can secrete cytokines and chemokines to rapidly recruit bone-marrow-derived monocytes into the lung, resulting in a considerable expansion of the macrophage pool [42,43]. Supply of bone marrow and spleen monocytes to circulation and then to the site of inflammation can maintain the number of Mo/Mφ relatively stable with a slight increase. Considering the dramatic expansion of IM, it is also likely that the local IM can proliferate following infection, as recently reported [9]. The decrease in AMs may be because (a) AMs undergo apoptosis and degradation in fighting with the pathogen as one of the primary host defence cells locally in the early stage of Chlamydia invasion and (b) AM transforms into IM to continue their fight with the pathogen in cooperation with other innate and adaptive immune cells.

The most significant finding in the study is the M1 polarisation of different macrophage subpopulations in the lung and the critical role of iNKT cells in directing this process following *chlamydial* infection. Several studies in vivo have suggested the impact of intracellular bacteria on the polarisation of macrophages. M1 macrophages can produce pro-inflammatory cytokines such as IFN-γ and TNF-α, directly destroying intracellular pathogens and promoting local Th1 cell responses [9,44]. Shirey et al. found that *Francisella tularensis* could induce M2 polarisation to survive at the expense of the host [45,46]. In contrast, our data show that *chlamydial* lung infection predominantly promoted the macrophage subpopulation towards M1 polarisation in vivo without significantly impacting M2 macrophages (Figure 3, Figure 4, Figure 5 and Figure 6). Preclinical studies using mouse models have demonstrated that the induction of Th1 responses, particularly IFN-γ production by T cells, is critical for protecting against *chlamydial* lung infections [47,48,49]. In addition to the most dramatic increase in IM in numbers (Figure 2), our data also show the biggest and the longest M1 polarisation in the IM subpopulation following *chlamydial* infection among the three analysed subpopulations. Therefore, it is likely that the IM is the most crucial macrophage subpopulation in host defence against *C. m*. lung infection. Based on the difference in the peak of M1 polarisation in the three subpopulations, IM is likely more benefited from Th1 cells than other subpopulations because its peak of M1 polarisation was on day 7 post-infection when T cell response became dominant [47]. At the same time, AM and Mo/Mφ showed the polarisation peak on day 3 post-infection when innate immune cells are dominantly activated. The production of IFN-γ by innate and adaptive immune cells may indirectly initiate and maintain the M1 polarisation of IM and other macrophage populations [50].

Our data show a broad, significant impact of iNKT cells on macrophage subpopulations, polarisation, cytokine production, and function. In particular, we found a substantial effect of iNKT cells on M1 polarisation in all the tested macrophage subpopulations. By analysing the four reported signalling pathways in the WT-Mφ and iNKT KO-Mφ, we found a close association of the JAK/STAT pathway with the high M1 polarisation in WT-Mφ (Figure 8). The JAK/STAT signalling pathway has been implicated in mediating the biologic responses induced by many cytokines [51]. Cytokines can use the JAK/STAT pathway to regulate the macrophage phenotype and function. There are four JAK family members (Jak1, Jak2, Jak3, and Tyk2), and seven STATs (STATs 1–6, including two STAT5 genes) have been identified in the JAK/STAT pathway [52]. Evidence suggests that IFN-γ is one of the most potent endogenous M1-macrophage-activating factors and uses the JAK/STAT signalling pathway [37,53]. IFN-γ can specifically activate the JAK1, JAK2, and STAT1 of the JAK/STAT pathway and further promote macrophages to produce iNOS and TNF-α (M1 phenotype) [51,52,53]. Our previous data show that iNKT cells promote NK cells and CD4/CD8 T cells to produce IFN-γ in the *C. m*. lung infection model [29,30,54]. The reduced level of IFN-γ production and the non-activated JAK/STAT pathway may be among the reasons that lead to reduced iNOS and TNF-α expression by iNKT KO-Mφ compared to WT-Mφ [55]. However, it should be pointed out that this study did not address how NKT cells modulate macrophage subsets and function. Cell–cell contact and cytokine production are likely involved, based on our findings on the interaction between NKTs and DCs. We have reported that the NK cell-related receptors on NKTs and the IFN-γ production by NKT cells play an essential role in the modulating function. We think a similar mechanism could exist in the NKT and macrophage interaction. Since NKTs can quickly produce large amounts of IFN-γ in the early stage of infection, it would not be surprising that NKTs use this mechanism to promote the polarisation of M1. Thus, it could have positive feedback for overall M1/Th1 responses, consequently enhancing infection control. A comprehensive study on the mechanism by which NKTs modulate macrophage function is a priority in this field.

iNKT cells can greatly promote M1 polarisation, but the M1 polarisation and macrophage responses to *chlamydial* infection are not entirely dependent on iNKT cells. Although the expansion of macrophages in the lungs of iNKT KO mice is significantly lower than in those of WT mice, iNKT KO mice still showed a mild increase in the macrophage population with a certain degree of M1 polarisation following *chlamydial* infection. Indeed, the absolute number of M1 cells in the iNKT KO mice increased after infection. The possible contribution of other cells or molecules in the modulation of macrophage function could explain why the recipient of iNKT KO-Mφ also showed a certain degree of protection (Figure 7).

Notably, the study presented here is not without limitations, and further study to address the limitations is needed to understand the modulating effect of iNKT on macrophages fully. For example, the NKT cells’ mechanism of modulating macrophages has hardly been tested. In particular, the possible direct and indirect interaction between NKTs and macrophages remains unclear. In addition, the data on signalling pathway analysis are only at the transcription (mRNA) level. Although the mRNA data strongly suggest the involvement of this pathway in the process, further study to confirm it at the protein level, including the phosphorylation status of these proteins, is necessary. Moreover, this study only examined macrophages in the NKT-deficient mice. A complementary approach to compensate isolated NKT cells from WT mice to the NKT-deficient mice would be essential to confirm the modulating effect of these cells on macrophages, as we reported in the study examining the influence of NKT cells on lung dendritic cells (LDC) [56]. Furthermore, it remains unclear if the iNKT cells promote LM polarisation through direct influence or indirectly through other cells such as NK cells, DCs, and T cells.

## 5. Conclusions

This study demonstrated a preferential increase in M1-type macrophages in the lungs of mice following chlamydial infection with higher expression of iNOS and TNF-α. However, the preferential shift towards M1 was much less in iNKT KO mice, suggesting a modulating role of iNKT on monocyte/macrophage function in protection. The modulating effect is confirmed by the adoptive transfer of LMs isolated from iNKT KO mice (iNKT KO-Mφ), which conferred less protection than those isolated from wild-type mice (WT-Mφ). This study also shows significantly reduced gene expression of the JAK/STAT signalling pathway molecules in iNKT KO-Mφ. The data imply an essential role of iNKT in promoting LM polarisation to the M1 direction, which is functionally relevant to host defence against *Chlamydiae*, an intracellular bacterial infection.

## Figures and Tables

**Figure 1 cells-13-00133-f001:**
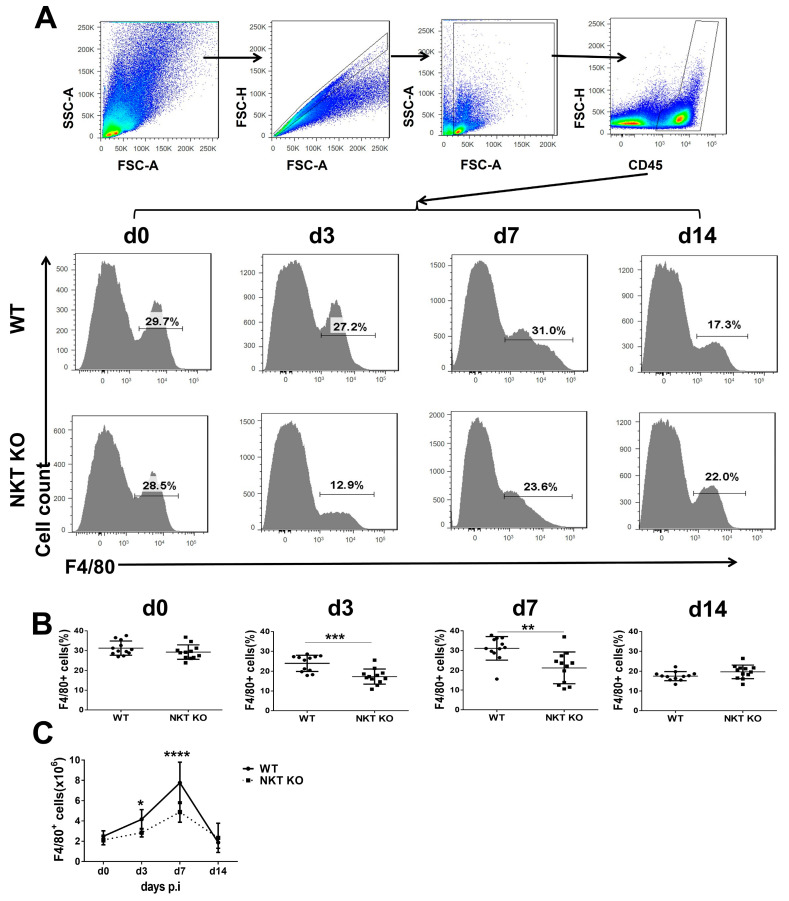
Reduced macrophage responses following lung *C. m.* in iNKT KO mice. WT or iNKT KO mice were inoculated intranasally with *C. m.* (1 × 10^3^ IFUs) and were killed on the indicated days. The mouse lungs were isolated, and the single lung cells were analysed by flow cytometry. (**A**) After the exclusion of doublets and debris, myeloid cells were gated by CD45 staining. The CD45-positive and F4/80-positive cells are identified as macrophages. The gating strategy (**top**) and the percentage of macrophages (CD45^+^F4/80^+^ cells) in the lung cells of each group at different time points of infection (**bottom**) are shown. (**B**) Summary data of flow cytometry analysis show the percentage of F4/80^+^ cells in CD45^+^ lung cells. (**C**) Following chlamydial lung infection, the absolute number of F4/80^+^ lung cells in iNKT KO and WT mice. Statistical analysis for B was performed using a nonparametric Mann–Whitney test. The statistical significance of C was determined using a two-way analysis of variance (ANOVA) with Sidak’s multiple comparisons. Three independent experiments with four mice in each group were performed, and data from 3 experiments were combined. *n* = 12 (WT) and 12 (NKT KO). Data are shown as the mean ± SD (* *p* < 0.1; ** *p* < 0.01; *** *p* < 0.001; **** *p* < 0.0001).

**Figure 2 cells-13-00133-f002:**
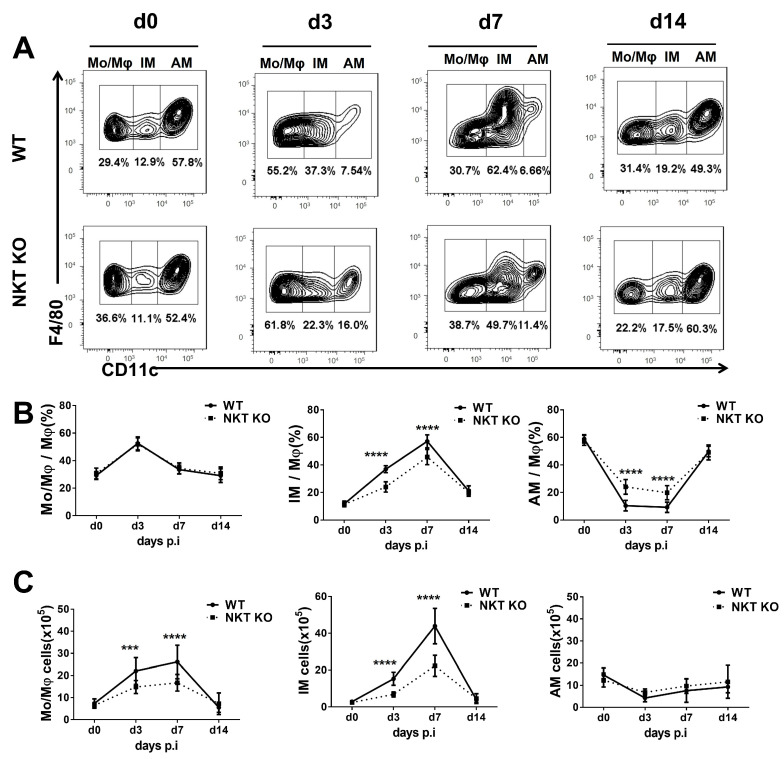
Kinetics of macrophage subpopulation changes during *C. m.* infection in WT and iNKT KO mice. Lung cells were prepared on days 0, 3, 7, and 14 post-infection, and the macrophage subpopulations were analysed by flow cytometry (F4/80^pos^CD11c^high^ for AMs, F4/80^pos^CD11c^low^ for IMs, and F4/80^pos^CD11c^neg^ for Mo/Mφ). (**A**) Representative flow cytometry graphs show Mo/Mφ, IMs, and AMs in the two groups at different times following infection. (**B**) Summary data of the percentages of Mo/Mφ, IMs, and AMs in WT mice and iNKT KO mice after *C. m.* lung infection. (**C**) The absolute number of lung macrophages of various subpopulations in WT or iNKT KO mice during *C. m.* infection. Statistical significance was determined using a two-way analysis of variance (ANOVA) with Sidak’s multiple comparisons. Three independent experiments with four mice in each group were performed. *n* = 12 from 3 independent experiments. Data are expressed as mean ± SD (*** *p* < 0.001; **** *p* < 0.0001).

**Figure 3 cells-13-00133-f003:**
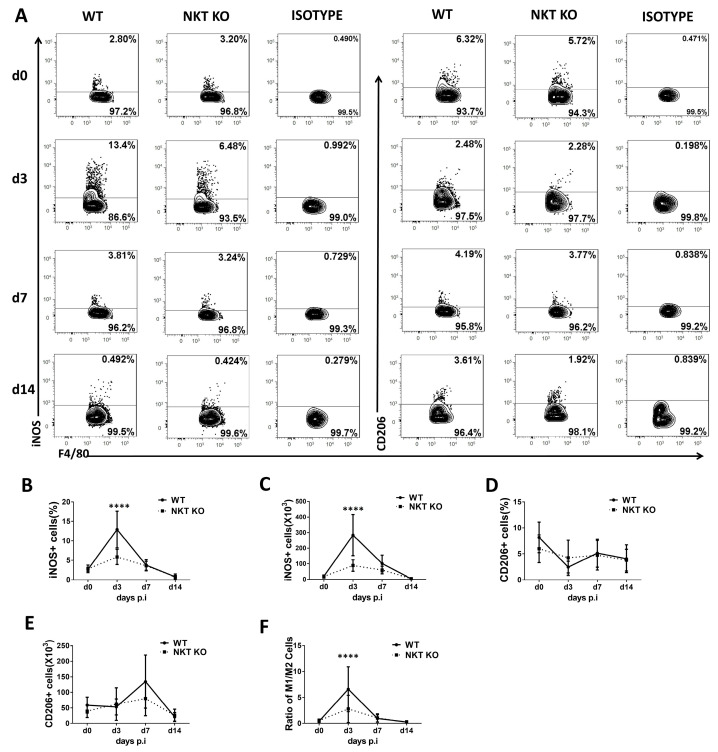
Polarisation status of Mo/Mφ in WT or iNKT KO mice during *C. m.* lung infection. WT and iNKT KO mice were treated with *C. m.* (1 × 10^3^ IFUs/mouse) and sacrificed on days 0, 3, 7, and 14 after *C. m.* infection. F4/80^pos^CD11c^neg^ lung cells were identified as Mo/Mφ. iNOS was selected as a marker for the M1 phenotype and CD206 for the M2 phenotype. (**A**) Representative flow cytometry graphs showing staining of iNOS+ or CD206+ Mo/Mφ cells. (**B**,**C**) Flow cytometry data were summarised to show the percentage (**B**) and absolute number (**C**) of iNOS+ Mo/Mφ cells in the different groups. (**D**,**E**) Flow cytometry data were summarised to show the percentage (**D**) and absolute number (**E**) of CD206+ Mo/Mφ cells in the two groups. (**F**) The M1/M2 cells ratio in Mo/Mφ during *C. m.* infection. Statistical significance was determined using a two-way analysis of variance (ANOVA) with Sidak’s multiple comparisons. Results represent three independent experiments (four mice in each group in each experiment). *n* = 12. Data are shown as the mean ± SD (**** *p* < 0.0001).

**Figure 4 cells-13-00133-f004:**
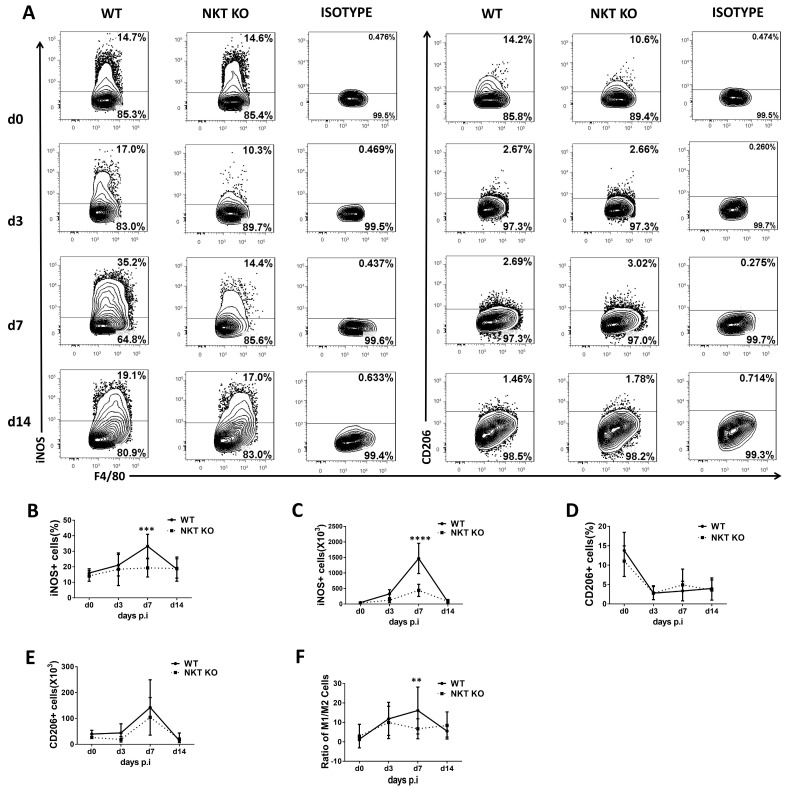
Polarisation status of IMs in WT or iNKT KO mice following *C. m.* lung infection. Mice and lung cells were treated as described in Figure 3. IM cells were identified as described in Figure 2. (**A**) Representative flow cytometry graphs showing each group’s staining of iNOS+ or CD206+ IMs cells. (**B**,**C**) Flow cytometry data were summarised to show the percentage (**B**) and absolute number (**C**) of iNOS+ IMs cells in the different groups. (**D**,**E**) Flow cytometry data were summarised to show the percentage (**D**) and absolute number (**E**) of CD206+ IMs cells in the two groups. (**F**) The ratio of M1/M2 cells in IMs during *C. m.* infection. Statistical significance was determined using a two-way analysis of variance (ANOVA) with Sidak’s multiple comparisons. At least three independent experiments with four mice in each group were performed, and *n* = 12 for three combined experiments. Data are shown as the mean ± SD (** *p* < 0.01; *** *p* < 0.001; **** *p* < 0.0001).

**Figure 5 cells-13-00133-f005:**
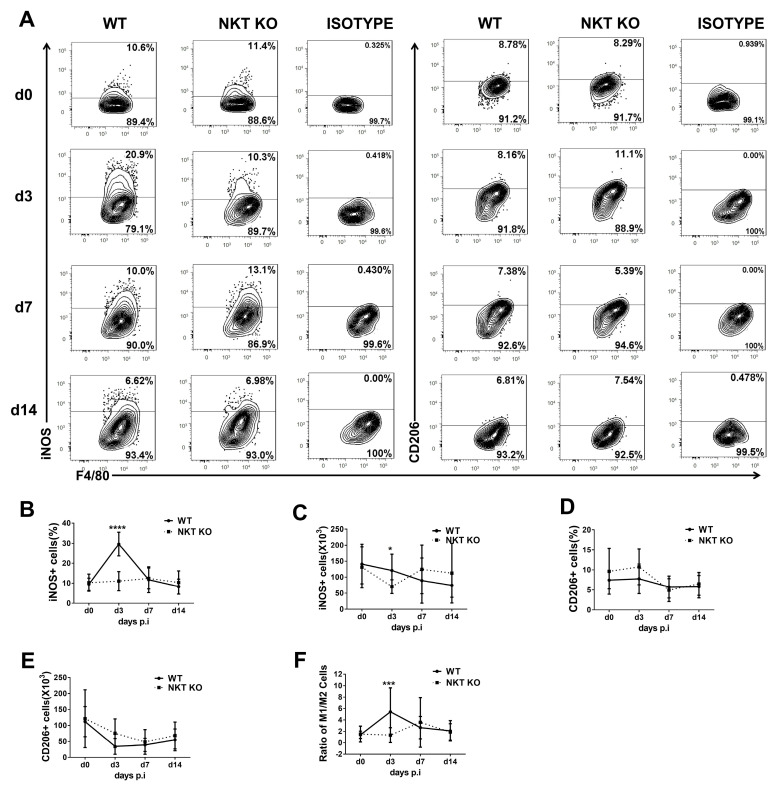
Polarisation status of AMs in WT or iNKT KO mice during *C. m.* lung infection. Mice and lung cells were treated as described in Figure 3. AM cells were identified as described in Figure 2. (**A**) Representative flow cytometry graphs showing each group’s staining of iNOS+ or CD206+ AMs cells. (**B**,**C**) Flow cytometry data were summarised to show the percentage (**B**) and absolute number (**C**) of iNOS+ AMs cells in the different groups. (**D**,**E**) Flow cytometry data were summarised to show the percentage (**D**) and absolute number (**E**) of CD206+ AMs cells in the two groups. (**F**) The ratio of M1/M2 cells in AMs during *C. m.* infection. Statistical significance was determined using a two-way analysis of variance (ANOVA) with Sidak’s multiple comparisons. Three independent experiments (four mice in each group in each experiment) with similar results were performed. *n* = 12. Data are shown as the mean ± SD (* *p* < 0.05; *** *p* < 0.001; **** *p* < 0.0001).

**Figure 6 cells-13-00133-f006:**
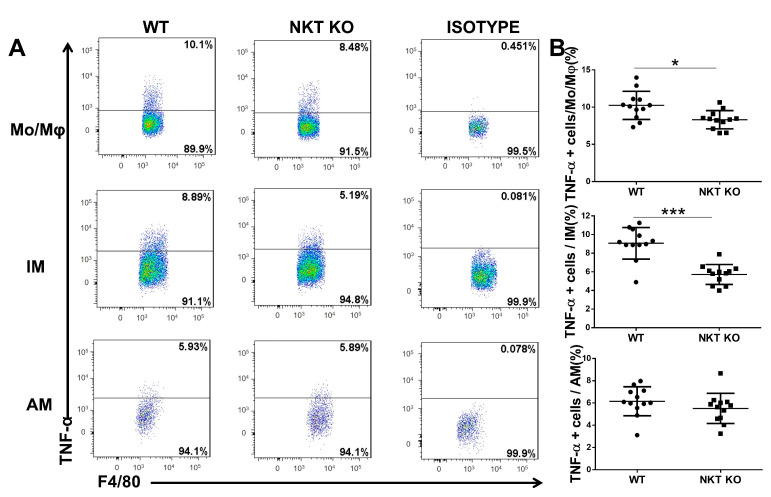
Fewer TNF-α + cells in AM, IM, and Mo/Mφ subpopulations of iNKT KO mice than WT mice following *C. m.* lung infection. Lung mononuclear cells were prepared on day 7 post-infection, and the macrophage subpopulations were pre-gated by flow cytometry (F4/80^pos^CD11c^high^ for AMs, F4/80^pos^CD11c^low^ for IMs, and F4/80^pos^CD11c^neg^ for Mo/Mφ). As described in Materials and Methods, the TNF-α-producing Mo/Mφ, IMs, and AMs were identified by intracellular cytokine staining. Representative flow cytometric plots (**A**) and a summary of the results (**B**) are shown. Results represent three independent experiments (four mice in each group in each experiment). *n* = 12. Statistical analysis for B was performed using a nonparametric Mann–Whitney test. Data are shown as the mean ± SD (* *p* < 0.05; *** *p* < 0.001).

**Figure 7 cells-13-00133-f007:**
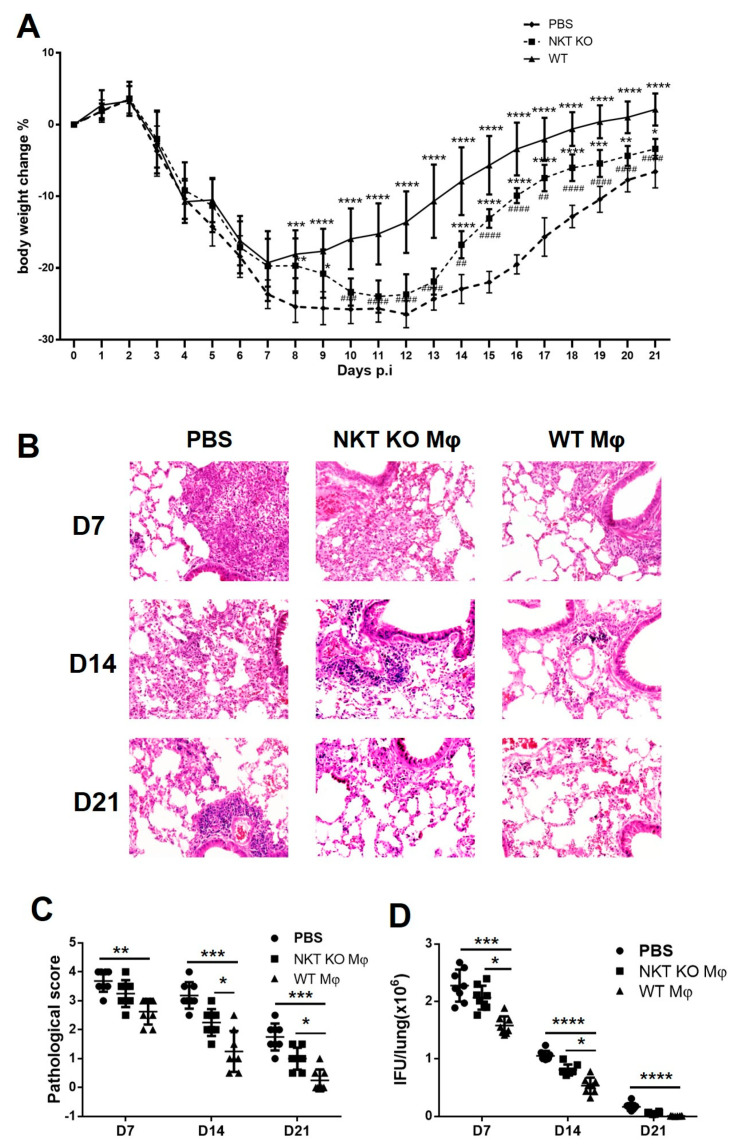
Adoptive transfer of WT-Mφ and iNKT KO-Mφ to test their function in vivo. Naive mice were intranasally transferred with WT-Mφ or iNKT KO-Mφ and challenged with *C. m.* Mice receiving PBS only with the same challenge infection were considered as controls. Mice were monitored daily for body weight changes and sacrificed at indicated days. (**A**) The percentage of body weight changes in the three groups of mice. Asterisks (*) on the WT Mφ line indicate the statistical significance of WT Mφ versus PBS, while asterisks on the iNKT KO Mφ line indicate the statistical significance of the iNKT KO Mφ versus PBS group. Pounds (## *p* < 0.01; ### *p* < 0.001; #### *p* < 0.0001) on the iNKT KO Mφ line indicate the statistical significance of WT Mφ versus iNKT KO Mφ. Statistical significance was determined using one-way ANOVA with repeated measurements for the bodyweight curve. (**B**) The lung tissue sections from different groups were routinely stained with H&E and analysed in ×400 magnification under light microscopy. (**C**) Semi-quantitative analysis of lung inflammation by pathological score as described in Materials and Methods. Statistical significance was determined using a Kruskal–Wallis test with Dunn’s post-test for histological scoring. (**D**) Lungs of different groups were homogenised and tested for chlamydial loads. Statistical analysis for chlamydial IFU in tissue was performed using a Kruskal–Wallis test with Dunn’s post-test. Data are shown as mean ± SD. Two independent experiments with 4 mice in each group were performed, and data from two experiments were combined (*n* = 8). Data are means ± SD (* *p* < 0.05; ** *p* < 0.01; *** *p* < 0.001; **** *p* < 0.0001).

**Figure 8 cells-13-00133-f008:**
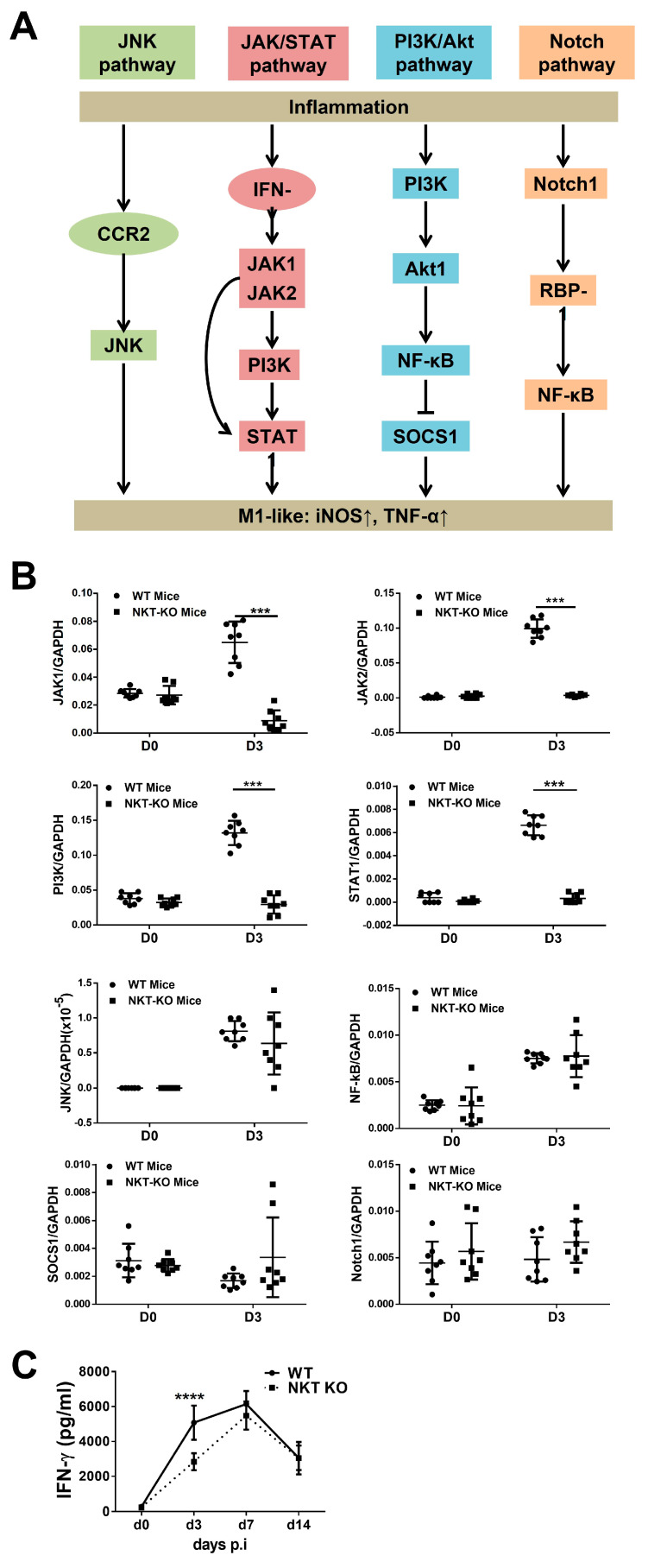
Detection of the JNK pathway, JAK/STAT pathway, PI3K/Akt pathway, and Notch pathway in macrophages by quantitative real-time RT-PCR. (**A**) Typical signalling molecules in the JNK, JAK/STAT, PI3K/Akt, and Notch signalling pathways are reportedly related to M1 polarisation. (**B**) WT or iNKT KO mice were infected intranasally with *C. m.* (1 × 10^3^ IFUs) and were sacrificed on the indicated days. The lungs were isolated, and the lung macrophages were separated by the MACS column. The mRNA levels of JAK1, JAK2, PI3K, STAT1, JNK, NF-κB, SOCS1, and Notch1 in lung macrophages were measured by real-time RT-PCR. For qPCR, the molecules were measured by the 2^−ΔΔCT^ method, and the data are shown as the ratios of each gene to the housekeeping gene (GAPDH), which is invariantly expressed in this chlamydial infection model [8]. Statistical significance for each group was determined using a nonparametric Mann–Whitney test. (**C**) ELISA assayed the protein levels of IFN-γ in lung homogenate. Statistical significance was determined using a two-way analysis of variance (ANOVA) with Sidak’s multiple comparisons. Results are representative of two independent experiments (four mice in each group in each experiment). *n* = 8. Data are shown as the mean ± SD (*** *p* < 0.001; **** *p* < 0.0001).

## Data Availability

Data was available by contacting authors by email.

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
