# Peer review of "Enhancement of Macrophage Immunity against Chlamydial Infection by Natural Killer T Cells"

_cells, 2024, doi:10.3390/cells13020133_

Round 1

Reviewer 1 Report

Comments and Suggestions for Authors

The paper concerns the macrophages during Chlamydia infection. Lung macrophage plays a vital role in host defense against bacterial infections. The topic taken up is interesting. No less results need additional experiments and clarifications.  

Quantitative real-time-PCR analysis

  • Two reference genes should be used for normalization. Normalization against a single reference gene is unacceptable unless the investigators present clear evidence confirming its invariant expression under the experimental conditions described.
  • There is no information on what quality the RNA was and how the quality of the RNA was determined (e.g., what RIN was)
  • There is no information on how cDNA was obtained;

Flow cytometry 

  • There is no information on whether the Authors determined extracellular or intracellular TNFα levels;
  • In line 352, there is “we examined the expression of TNFα.” Was it gene expression or maybe the amount of protein? 

Detection of the JNK pathway, JAK/STAT pathway, PI3K/Akt pathway and Notch pathway

The Authors measured only the gene expression of singular genes. To investigate signaling pathways, the Authors should determine the amount of proteins and their activity. There are unique panels for studying whole pathways. 

Others

There are editorial errors and colloquial language (e.g., our lab found)

Graphs are not self-explanatory, e.g., Figure 1B (% in comparison to which cell, total?) 

Comments on the Quality of English Language

There are editorial errors and colloquial language (e.g., our lab found)

Author Response

The paper concerns the macrophages during Chlamydia infection. Lung macrophage plays a vital role in host defense against bacterial infections. The topic taken up is interesting. No less results need additional experiments and clarifications.  

Quantitative real-time-PCR analysis

  • Two reference genes should be used for normalization. Normalization against a single reference gene is unacceptable unless the investigators present clear evidence confirming its invariant expression under the experimental conditions described.

Reply: We agree that two references would be better for normalization. Although we only used GAPDH as a reference gene in Quantitative real-time-PCR analysis, we did confirm that GAPDH is an invariantly expressed during chlamydia lung infection in our previous report (reference 8). Based the reviewer’s comment, we cited this paper in the legend of Figure 8 of the revision (line 448).

  • There is no information on what quality the RNA was and how the quality of the RNA was determined (e.g., what RIN was)

Reply: Due to limitations in instrument availability, we didn’t use Agilent Bioanalyzer for RIN assessment. Instead, we use Nanodrop to measure OD260/280 for all RNA samples as a quality control method. This is explained in lines 181-182.

There is no information on how cDNA was obtained;

Reply: We added this information in lines 183-184.

Flow cytometry 

  • There is no information on whether the Authors determined extracellular or intracellular TNFα levels;

Reply: We detected intracellular TNFα levels. The details is in the section of Method 2.5 on flow cytometry analysis (lines 154-156). Cells were stained with anti-TNF-α-PE mAbs after being fixed and washed in a permeabilization buffer.

  • In line 352, there is “we examined the expression of TNFα.” Was it gene expression or maybe the amount of protein? 

Reply: Sorry for the confusion. We have changed “expression” into “amount of TNF protein” (line 356-357).

Detection of the JNK pathway, JAK/STAT pathway, PI3K/Akt pathway and Notch pathway

The Authors measured only the gene expression of singular genes. To investigate signaling pathways, the Authors should determine the amount of proteins and their activity. There are unique panels for studying whole pathways. 

Reply: Thanks for the suggestion. we agree that it will provide stronger evidence if we have data on the protein levels of the pathway‐related molecules and phosphorylation status. Unfortunately, due to breeding issues and difficulty getting the KO mice in our animal facility at present time, we are unable to do this measurement for now. On the other hand, since the major goal of the present study focusing on the examination of LM subpopulations in chlamydial infection and the influence of NKT on LM. The gene expression data on signaling pathways here only provide a direction for further mechanistic studies. We feel a comprehensive signaling pathways study is needed to elucidate this mechanism, including a systemic investigation of gene expression, protein production and molecule phosphorylation in in vivo and in vitro conditions. This will need an independent investigation based on the amount of work and the significance of the related information. Considering the lack of comprehensive understanding of the signaling pathway at this stage, we used “likely” to soften the tone of this finding in the Abstract and pointed out the limitation of the available data in the Discussion (571-572), particularly on the lack of the protein data as commented by the reviewer.

Others

There are editorial errors and colloquial language (e.g., our lab found)

Reply: We have carefully rechecked and corrected the language in the revision. In particular, we changed “our lab found” into “we found”. Thanks!

Graphs are not self-explanatory, e.g., Figure 1B (% in comparison to which cell, total?) 

Reply: According to your suggestion, we have added explanation for Figure 1 and modified other figure legends for clarification.

Reviewer 2 Report

Comments and Suggestions for Authors

Comments to the Authors:

The authors effectively presented their findings, contributing significantly to the understanding of immunity mediated by innate immune cells like macrophages in respiratory diseases. Despite this, some details are missing, and I offer the following comments for your consideration:

The reduction of AM after infection is noted, but the significance of this reduction in terms of pathogen clearance is unclear. Have you investigated the resolution of inflammation during the time course?

In Fig. 3, titled "Polarized status of Mo/Mφ in WT or iNKT KO," it would be helpful to understand how you tracked the polarized status of Mo/Mφ macrophages. Additionally, on page 8 (lines 298-300), could you clarify what is meant by "M1-like Mo/Mφ"?

The study heavily focuses on NKT KO, but the mechanism through which iNKTs regulate lung macrophage subpopulations is not discussed. Please address this gap and explicitly mention the study's limitations.

During the transition of IM to AM, the manuscript emphasizes the significant M1 shift. However, there is no mention of significant changes in M2 macrophages. Please provide clarification or discuss any observations related to M2 macrophages during this transition.

Overall, the manuscript holds promise but addressing these concerns will enhance its clarity.

Author Response

The authors effectively presented their findings, contributing significantly to the understanding of immunity mediated by innate immune cells like macrophages in respiratory diseases. Despite this, some details are missing, and I offer the following comments for your consideration:

The reduction of AM after infection is noted, but the significance of this reduction in terms of pathogen clearance is unclear. Have you investigated the resolution of inflammation during the time course?

Reply: We have indeed investigated the resolution of inflammation during the time course. Through about 30 years of study on this mouse model of Chlamydial lung infection, we have observed and reported that on the seventh-day post-infection, the bacterial load in the mouse lungs is close to its peak, with histopathological evidence of inflammatory infiltration covering nearly 80% of the area. By the fourteenth day post-infection, the bacterial load in the mouse lungs reduced to minimal level and the inflammatory infiltration area was reduced to less than 30%. Therefore, the reduction in alveolar macrophage is in line with the dynamic resolution of inflammation. To make this to be clear to the readers, we added a sentence to clarify this and cited our previous work (line 231-232).

In Fig. 3, titled "Polarized status of Mo/Mφ in WT or iNKT KO," it would be helpful to understand how you tracked the polarized status of Mo/Mφ macrophages. Additionally, on page 8 (lines 298-300), could you clarify what is meant by "M1-like Mo/Mφ"?

Reply: In Fig. 3, we tracked the polarization status of Mo/Mφ macrophages by assessing specific markers associated with M1/M2 polarization, meaning iNOS(M1) /CD206 (M2) expression. This is mentioned in the legend as “iNOS was selected as a marker for the M1 phenotype and CD206 for the M2 phenotype”. We also specified this in line 301-304. Similarly, the term "M1-like Mo/Mφ" on page 8 refers to Mo/Mφ macrophages displaying characteristic M1 marker, particularly the heightened expression of pro-inflammatory iNOS.

The study heavily focuses on NKT KO, but the mechanism through which iNKTs regulate lung macrophage subpopulations is not discussed. Please address this gap and explicitly mention the study's limitations.

Reply: Thanks for the comments and suggestions. Based on the suggestion. We have added sentences in two paragraphs of the Discussion. First, we had a discussion on the possible mechanisms by which NKT modulates macrophage function and subpopulation (Line 545-555). Second, we add sentences in the last paragraph pointing out that the lack of mechanism data for the modulating effect is a limitation and stress the importance of this type of study in the future (566-570).

During the transition of IM to AM, the manuscript emphasizes the significant M1 shift. However, there is no mention of significant changes in M2 macrophages. Please provide clarification or discuss any observations related to M2 macrophages during this transition.

Reply: The primary goal of the present study is to examine the influence of NKT on LM, particularly on various LM subpopulations in chlamydial infection. However, based on the findings from Figures 3, 4, and 5, we did not observe significant impact of NKT cells on the percentage and absolute number of M2 macrophages during Chlamydia infection. This is why we did not talk much on M2 macrophages in the original Discussion. We did mention in the results part that “the absolute number of CD206+ macrophages (M2-like) in all the subpopulations had no significant difference between WT and iNKT KO mice at the tested various time points, representing early and late stages of infection” (line 305-308). Nevertheless, we agree with the reviewer it would be helpful for the reader if we address the point in the Discussion. Therefore, we added sentences in the second paragraph to clarify this finding and added comments on the implication of this finding (line 475-477).

Overall, the manuscript holds promise but addressing these concerns will enhance its clarity.